# A Systematic Review of Literature on User Behavior in Video Game Live Streaming

**DOI:** 10.3390/ijerph17093328

**Published:** 2020-05-11

**Authors:** Yi Li, Chongli Wang, Jing Liu

**Affiliations:** 1School of Economics and Management, Chongqing University of Posts and Telecommunications, Chongqing 400065, China; liyi@cqupt.edu.cn; 2Information and Library Center, Chongqing Medical and Pharmaceutical College, Chongqing 401331, China; joanna_lj01@163.com

**Keywords:** video game live streaming, streamer demand, audience demand, interactive behavior, platform

## Abstract

Video game live streaming is a kind of real-time video social media that integrates traditional broadcasting and online gaming. With the rapid popularity of video game live streaming in the past decade, researchers have started to investigate the relationship between the use of video game live streaming and various psychological variables. In order to fully understand the factors that affected user participation (streamers and audiences) in video game live streaming and provide a reference to the mental health issues of Internet addiction, this paper summarizes the relevant literature on user behavior in video game live streaming. First, we comprehensively searched literature in six social science databases and thus obtained 24 papers that meet our inclusion criteria. Second, the above literature was presented in table form for classification and we found that the effect factors of user behavior in video game live streaming mainly include user demands and platform impact. Based on Use and Satisfaction theory, this paper reviewed the following four aspects: streamer demand, audience demand, interaction behavior and platform impact, then a relevant theoretical framework was constructed. Finally, this paper looks forward to possible future research topics based on the research platform, research data and research content and so on, hoping to provide a foundation and new ideas for future research.

## 1. Introduction

Video game live streaming is a kind of real-time video social media that integrates traditional broadcasting and online gaming [1,2]. In recent years, the online broadcast industry has developed rapidly with the emergence of various live streaming platforms, such as Twitch, YouTube, Douyu, Huya, and so on. Twitch (www.twitch.tv) and YouTube Gaming (gaming.youtube.com) had more than 470 million regular visitor members in 2016 (more than 50% of gamers in the US, Europe, and Asia Pacific) [3]. By 2018, at least 1 million people were watching live streaming on Twitch at any time, and by the end of the year the total duration of viewing time on the platform was recorded at 434 billion minutes. In addition, the number of streamers on Twitch increased from 2 million in 2017 to 3 million in 2018, with nearly 500,000 streamers visiting this platform daily [4]. The addiction to video game live streaming has also raised concerns. In the area of mental health, Internet addiction has been described as a disorder in which individuals use the Internet excessively [5]. Addiction to online gaming has been included separately in the fifth edition of Diagnostic and Statistical Manual of Mental Disorders as a tentative disorder [6]. Problematic use of online gaming has become a popular topic of academic concern. However, the relevant research is scattered between video game live streaming use and various psychological variables.

Video game live streaming users can be divided into streamers and audiences. The main difference between them is that the former produce information or content, while the latter receives or consumes it [7]. The streamers display game skills through live streaming and audiences can watch and learn the game through live streaming. The two parties can interact by communicating, rewarding, subscribing, and so on. Accordingly, studies on user behavior are summarized in three aspects: effect factor on streamer participation in live streaming, effect factors on audience watching live streaming, and effect factors on their interactive behavior (chat, reward, subscription, etc.). Zhao et al. [8] argued that performance expectation from the platform and perceived website attractiveness could significantly promote streamers’ willingness to continue streaming. Based on the theory of “Use and Satisfaction”, Sjöblom and Hamari [1] divided audience demands to watch live streaming into five categories: cognition, emotion, personal integration, social integration, and stress release. Gros et al. [9] found that the audience rewards intention was influenced by the number of viewers and watching duration. In addition, this paper believes that the environmental factor plays an important role in the behavior of live streaming users. Therefore, the category of “platform impact” is added to the previous classification.

In summary, the literature was reviewed from the following four aspects: streamer demand, audience demand, user interaction behavior, and platform impact; and the relevant theoretical framework was constructed. Finally, the paper looks ahead at the future research directions. Through this literature review, we hope to provide a clearer picture of the current research state of the factors, which impacted user participation in video game live streaming. Also, we hope to provide a reference for the psychology of addiction to video game live streaming.

## 2. Literature Retrieval

Literature retrieval refers to the Preferred Reporting Items for Systematic Reviews and Meta-Analyses (PRISMA) guidelines [10]. The proceeding sections outline this methodology, beginning with a description of the search strategy, followed by the inclusion criteria and finally the literature collection results.

### 2.1. The Search Strategy

This paper adopts the following steps to retrieve and collect existing literature on video game live streaming. First, through literature search, we gathered various expressions of video game live streaming and determined the retrieval keywords: “live streaming,” “Internet broadcast,” “network broadcast,” “webcast,” “video game,” “online game,”. Thereafter, various combinations of these keywords were created. Next, we selected the most commonly used social science databases: Elsevier SD, EBSCO, Spring link, Emerald, Wiley Online Library, and Google Scholar. This paper takes into account that the earlier video game live streaming platform Twitch was officially founded in 2011, after which live gaming was widely used. Meanwhile most of academic researches on video game live streaming have occurred from 2012 onwards. Therefore, the search time was determined to be limited to peer-reviewed articles in English published between January 2012 and September 2019. Then, for the retrieved literature, we traced relevant studies and the referenced studies. Furthermore, a secondary search was conducted in April of 2020 to locate any additional articles published since the original search.

In adherence to the PRISMA guidelines [10] (see Figure 1), articles obtained through the database searches were first screened based on their titles and keywords, then abstracts, and finally the entire journal article was considered. At the initial title and key words screening stage, articles were included if the title and key word indicated that the research pertained to use behavior in live streaming and online gaming. Then, at the abstract screening stage, articles were included if abstract indicated that the research pertained to user behavior and some aspect of impact factor. Finally, the full-text articles of the accepted abstracts were screened for eligibility; articles were rejected at this stage if article could not be obtained, and/or only studied live streaming user behavior, and/or only studied the behavior of online game users, and/or had no clear user subject, and/or had no clear research data and available methods.

### 2.2. The Inclusion Criteria

To ensure rigidity of this literature review, studies were required to meet the following inclusion criteria:Research content—the main matter of the literature should be about user behavior and its effect factors on video game live streaming, including when using platforms such as Twitch, YouTube Gaming, DouYu TV, Huya.com, as well as other popular online game platforms. And the article must include clear live streaming user objects, research methods, research data.Date must be from January 2012 to April 2020.Article must have been written in English.Article must either be a peer-reviewed journal or a conference article.

### 2.3. The Literature Collection Results

A total of 24 studies were selected. These examined three main areas: effect factors on streamer participation (six papers), effect factors on audience participation (twelvepapers), and effect factors on interactive behaviors (eight papers). Notably, two of the selected twelve papers examining the effect factors of audience participation also mentioned interactive behavior. Table A1, Table A2 and Table A3 in Appendix A present the distribution/classification of these studies.

Sorting through the existing literature revealed that Kaytoue et al. [11] were early scholars who studied video game live streaming through in-depth data analysis. Through crawling and analyzing relevant data of more than “100 days” on the Twitch platform, they found that, for the audience to watch video game live streaming, effect factors included the popularity of the game, gaming skills of the streamer, and whether the streamer is a professional player. Second, Twitch was a commonly used research platform although some scholars also conducted research on YouTube, YouNow, Douyu, Huya, and other such platforms. Third, the theory of “Use and Satisfaction” was a commonly used theoretical basis; a quarter of the obtained literature explicitly used this theory to study video game live streaming user behavior. Fourth, empirical research was a common research method, which collected data through questionnaire surveys, network data crawling, interviews, and other methods; some scholars also conducted research through laboratory experiments.

This paper divided the main issues into four types: streamer demand, audience demand, user interaction behavior, and platform influence. This classification referred to studies by Scheibe [12] and Zimmer [7], who divided user behavior into three categories: the streamers who produce videos; the audience that watches the videos; the chats and rewards between the two groups. This paper believed that the environmental factor played an important role in the behavior of live streaming users. Therefore, the category of “platform impact” was added to the previous classification. These four main topics would be highlighted in the following text, and on the basis of these issues the research framework of video game live streaming could be built. This is presented in Figure 2.

## 3. Concept of Video Game Live Streaming

### 3.1. Definition of Video Game Live Streaming

The existing literature mostly defines video game live streaming based on form and content. With regard to form, video game live streaming is described as a new media that integrates traditional broadcast and online gaming [1,2], such as on Twitch, Huya, etc. As with traditional broadcasts, video game live streaming retains the form of one-way transmission, but compared with the former, the consumption experience of video game live streaming is more positive. In the online gaming environment, video game live streaming retains multi-directional interaction, but is more passive than playing online games [1]. In addition, audiences and streamers can chat in real time during video game live streaming, which would form a community around streamer [13]. Audiences in this community can interact with their favorite celebrities (streamers), and this will generate a sense of intimacy; few other media can match that [14]. Pellicone and Ahn [2] also agreed that, along with video games, “intimacy, celebrity, community, content creation, and consumption” were other elements that were interwoven in the video game live streaming. Therefore, video game live streaming is a form of media integrating the public, communities, interaction, and passivity, while bridging the gap between online games and traditional video media (such as TV).

In terms of content, video game live streaming is defined as a network broadcast with “online games” as the specific content [12,15], such as Dailymotion Games and Douyu. Pellicone and Ahn [2] considered live streaming to be a kind of performance—to spread online gaming culture. On the live streaming game platform, the behavior of streamers and audiences is based on the “online games”; that is, streamers can make and share videos in real time as they play online games, while simultaneously communicate with their game-specific audience [9]. In addition, audiences can also watch videos related to particular “online games” on the platform, search information related to those online games, learn online gaming skills, and so on.

### 3.2. Users of Video Game Live Streaming

Video game live streaming users are divided into two groups, streamer and the audience, and the differences between them are related to information production and reception [7]. Streamers are the producers of live streaming. They record and publish real-time video [12] with their own mobile devices (such as smart phones, tablets, and desktop computers) [7,15,16]. As aforementioned, streamers can display game skills, teach game strategies, and manage their self-image by chatting with audiences on the platform [17], such as expressing their interests and attitudes towards life. Audiences are consumers of live streaming and they watch the content on mobile devices or desktop computers. Live streaming platforms enable audiences to chat with streamers and other consumers, reward streamers, subscribe to the live streaming channels [7,15], as well as acquire information and learn skills related to that game [18]. Therefore, when considering users who are addicted to live game play, streamers and audiences should be considered separately.

### 3.3. Features of Video Game Live Streaming

There are three main features of video game live streaming mentioned in the existing literature:

First, live streaming is performed in real-time; that is, streamers share online videos of playing game [19] while audiences watch those same videos concurrently. In other words, production and consumption of real-time videos are completed at the same time [7]. In addition, streamers and audiences can also communicate with each other simultaneously. Generally, the platform will display the screen of the streamer playing games and chatting with the audience in real time [18]. Therefore, video game live streaming is a real-time video media.

Second, the activity is highly conducive to sociability; that is, live streaming is a type of social media [7,17,19]. Similarly, video game live streaming is also social. Hamilton et al. [19] explained that social interaction was the experience of participating in major activities. Therefore, these studies situated Twitch communities in line with McMillan and Chavis’ concept of the “sense of community,” which comprised the four aspects of “membership, influence, fulfillment of demands, and emotional connection” [20]. In view of this, this paper regards user interaction in live streaming as a social activity. Streamers and audiences can interact by chatting, giving rewards, subscribing, and learning online game skills from each other [21]. In addition, Fietkiewicz and Stock [15] also proposed that the design of the gamification mechanism on video game live streaming platform strengthened the interaction between streamers and audiences, such as rewards, ratings, badges, and other gamification mechanisms.

The third feature is the dissemination of suspense; that is, compared to traditional media (in which most of videos are recorded in advance, such as on iQIYI and Tencent Video), video game live streaming has the characteristics of suspense. For example, online games and other activities with a competitive nature encapsulate high moments, low moments, and a suspenseful result, whereas when watching pre-recorded media, the result is fait accompli, making content less engaging because of spoilers or a pre-determined outcome [22]. Wulf et al. [23] agreed that compared to watching pre-recorded video (such as a TV series or a movie), audiences might be affected by the excitement and suspense while hoping the streamer win the game or complete the challenge.

It has been mentioned in existing literature that there are three main features of video game live streaming: real time, social media and suspense compared to other video platforms and social platforms. Therefore, these three characteristics should be fully taken into account when investigating the factors that affect user behavior in video game live streaming.

### 3.4. Community Classification of Video Game Live Streaming

According to different types of games, communities of video game live streaming can be divided into three categories: Let’s Players (self-entertainment), Speedrunner (quickly complete a game), and E-sports (professional competition) [24,25]. Among them, Smith et al. [24] held that Speedrunner communities were relatively special. In such communities, streamers not only exhibit simple game skills but also teach game strategies to the audience by taking advantage of bugs found on the game platform. In addition, among these three types of communities, E-sports communities were the most popular for audiences, while Speedrunner communities and Let’s Players communities were on the rise [24]. However, Churchill and Xu [14] found that the largest number of live streamers were in Let’s Players communities, and the lowest in e-sports communities. To summarize, the audiences’ favorite community for video game live streaming is different from streamers. Different community characteristics can meet the different demands of streamers and audiences. Therefore, the respective demands of streamers and audiences are discussed in the distribution of chapters 4 and 5 of this paper.

## 4. Streamer Demand

Table A1 presents the effect factors, research theories, research methods, and sample data from the six literature that impacting video game live streamer participation. In these six papers, the underlying theories of the studies are unclear except for one where the use of self-determination theory was mentioned. The indicated that the theoretical foundation of the streamer behavior is not solid enough. Second, methods of these studied involved questionnaires, interviews, and data mining. Effect factors mentioned in each paper were different and effect factors were more fragmented due to the small number of relevant studies. Therefore, this chapter mainly sorts effect factors on streaming or continuously streaming participation from a social integration, personal integration, and affection. This chapter also refers to use and satisfaction theory that media can meet cognition, affection, personal integration, social integration, and tension release demand of users [26]. In addition, the strength of monetary incentive among effect factors is controversial. Especially for professional streamers, the monetary incentive is an indispensable effect factor.

Social integration was a demand to strengthen ties with relatives, friends, and family [26]. According to Sjoblom and Hamari [1], this also included the demand to enhance connections with strangers, which could be realized through network chat, email communication, and so on. It mainly referred to the social demand of connection between streamers and audiences. In other words, the interaction between audiences and streamers would affect the willingness of both parties to participate in live streaming [2,8]. Hamilton et al. [19] took the Twitch platform as an example to explain that video game live streaming formed a virtual third informal community, and the audience size of this community varied from less than fifty people to thousands of people. More so, the platform gave streamers a sense of community [14], enabling them to fulfill their social integration demand, thus affecting their behavior of participation in live streaming. In interviews with streamers, some streamers mentioned that they would be more willing to put in a lot of time on the video game live streaming because one-third of audiences were regulars and had a friendly relationship with these regulars [19]. It is universally recognized that as a category of social platforms, live streaming platforms can meet the social demands of streamers by interaction with audience. Therefore, social integration should be taken into account when studying streamer addiction to video game live streaming.

Personal integration was a demand to improve personal confidence, credibility, and status [26], and mainly involved the fulfillment of self-presentation, self-realization, and a sense of accomplishment (i.e., reputation, status). The awareness of self-presentation played an important role in the behavior of live-streaming participants [2]; in front of audiences, some streamers often treat game play as a form of expression [19]. Tang et al. [17] explained that live streaming provided an instant opportunity and solid technical support for self-expression, and streamers could express their interests and showcase their talents in real time. In addition, through interviews with game streamers, Johnson and Woodcock [27] proposed that the sense of achievement had a positive impact on streamers’ participation in live streaming, since the reputation and status gained by them could satisfy their inner self-esteem. Half of the six studies in the collected literature were a personal integration demand. The research methods involved in these studies were questionnaires and platform data mining. The SP platform and Twitch platform were investigated on research platform.

Affection demand is an emotional, pleasurable and aesthetic experience [26]. That is, the demand to enjoy tasks, provide entertainment, share interests and hobbies, and seek challenges. First, the degree of task enjoyment would affect the willingness of a live streamer to participate in live streaming. Enjoying task is a kind of motivational guidance to express and entertain oneself through task performance, which focuses more on the emotional experience of the audience [8]. Gandolfi [28] showed in his research that the motivation of streamers to seek pleasure and entertainment positively impacted the behavior of streamers to participate in live streaming. Entertainment is not only limited to satisfaction when playing games, but also includes bringing happiness to the audience. In addition, Gandolfi [28] also found that the motivation of streamers to seek challenges would affect their behaviors of participating in live streaming. Therefore, challenges in video game live streaming were conducive to attracting streamers. Zhao et al. [8] found that fulfilling streamers’ affection demand have a positive impact on continuously streaming. Existing studies have been empirically analyzed in terms of enjoy tasks, provide entertainment, share interests and hobbies, seek challenges, and so on. There is no dispute that affective demand is positively relative to streaming. Therefore, when examining the reasons why streamers are addicted to video game live streaming, the affection demand should be fully considered.

In addition to the main factors of the aforementioned three demands, external rewards are also factors on participating in live streaming. External rewards are divided into reward measures of the platform (i.e., gamification mechanism) and rewards from the audience (i.e., rewards, subscriptions). Research showed that external rewards had positive effects on game players to continuously streaming [8,19,27]. For example, streamers would be motivated by monetary rewards and continued to stream. Furthermore, professional game streamers appeared because of the existence of external rewards. Specifically, monetary rewards were considered as salary. Hence, money encouraged streamers to continue streaming [19]. However, Johnson and Woodcock [27] found that professional streamer has felt precariousness about their long-term streaming careers, so whether the incentive effect of money diminishes in the face of such anxiety is worth investigating.

## 5. Audience Demand

Table A2 presents the effect factors, research theories, research methods, and sample data from twelve literature that studied factor affecting audience participation. Compared to investigating the effect factor of streamer participation and interactive behavior, there were a larger literature on audience participation in live streaming. There were three theories explicitly mentioned in these literatures, uses and gratifications theory, social identification theory and social cognition theory. In addition, three-quarters of the literature were studied by questionnaires; the rest used interviews and live streaming platform data mining. This section mainly analyzes the effect factors for audiences of live streaming from five aspects: social integration, personal integration, tension release, cognitive, and affective.

Social integration mainly refers to the social demand of the audience, including the connection between audiences and streamers, the audience and audience groups (other audiences). First, social interaction (with other audiences) was an important factor that impact audiences watching live streaming [29]. Hu et al. [30] discussed the reasons through a questionnaire survey and found that audiences’ identification with audience groups were positively associated with their intention to continue watching. When the audience realizes that values and beliefs of the audience group are in line with their own, she or he will enhance their sense of identity and attachment to the audience group and choose to maintain this connection through continued viewing. Second, the audience’s viewing intention was influenced by the relationship between the audience and the streamer [23]. Some scholars had offered relevant explanations. Hu et al. [30] believed that it was the role model effect, in which audiences worshiped and admired streamers because of their attitudes, values, and special skill, and thus audiences were willing to remain loyal and maintain a longer relationship. For example, the aggressiveness of streamers had a positive impact on the frequency of audience members watching their live stream [31]. The inherent personal brand effect and the unique identity image of the streamer would attract the audience [17]. Chen and Pan [32] took Twitch platform users’ chat comments as the experimental sample, and through methods of text mining and feature selection found that the interaction between the audience and the streamer was also an important factor affecting the number of audience members. Moreover, audience members were willing to spend more time on live streaming platforms in order to meet social demand [9,33]. Additionally, the more money spent on the platform, the stronger the audience member’s social motivation was [10]. Therefore, existing research has demonstrated that the social integration demand of the audience could be met in video game live streaming from multiple aspects such as audience identification, role model effect, and monetary investment.

Personal integration mainly comprises the audience’s self-presentation and self-recognition demand. Sjoblom et al. [34] found that audiences were more willing to continuously watch live streaming in order to show that they were active users and followers of social media. This also reflected their social self-awareness. If audiences wanted to gain self-recognition, they tended to watch live streaming with fewer participants in the audience [1]. Furthermore, the type of online game could impact on personal integration. The action games had a positive impact on audience satisfaction, while Real-Time Strategy (RTS) games had a negative impact on audience satisfaction [35]. Studies have shown that audiences with self-presentation demands were more likely to watch the live streaming consistently [35]. Therefore, we should also consider self-integration demand as a contributing factor to addiction. However, there is currently limited research on personal integration to effect audience participation in live streaming, which needs further explored in the future.

Tension release is inferred a demand for escape and diversion of attention [26]. That is, the demand to release stress and escape from reality. On the basis of the theory of use and satisfaction, Sjoblom and Hamari [1] found that releasing pressure and escaping from reality were important factors in audiences’ use of live streaming. Audiences who want to release pressure by watching video game live streaming were more likely to watch in casual and competitive communities [35]. They enjoyed the experience of watching and imagining the action. In addition, the degree of audience escapism positively correlated with the frequency of watching video streaming live streaming [31]. Watching live video game streaming distracts the audience from their daily activities and reduces their immersion in real life, hence, meeting their psychological demand of escapism.

The cognitive aspect infers to satisfying a demand for acquiring knowledge, information, and understanding [26]. That is, the demand for learning game skills and information acquisition. As aforementioned, some audiences watched video game live streaming in order to learn game skills and obtain relevant information [9]. For the information-oriented audiences, information seeking was observed to exert a moderating effect on the relationship between stress and negative outcomes in the video game live streaming [36]. Considering Twitch as an example, game skills could be acquired by watching video game live streaming, and relevant information could be obtained by viewing chat records through the platform search function. According to Sjoblom and Hamari [1], audience members with information acquisition needs were more inclined to watch video game live streaming of a specified content (such as teaching), because this could provide them with the desired information, however, this kind of audience paid equal attention to video game live streaming as others. For the study of cognitive demands, the existing literature is based on the use and satisfaction theory, using a questionnaire approach to explore. The video game live streaming can meet people’s cognitive demands, especially the game skills. But Sjöblom et al. [35] observed that live streaming with the main purpose of disseminating information or teaching something did not seem to attract substantive gratifications. Therefore, we thought that cognition might be not have a positive effect on indulging in video game live streaming.

Affective demand mainly includes suspense, excitement, entertainment, and interest. Lim et al. [37] thought that emotional engagement had indirect effects on repeating viewing live streaming. Wulf et al. [23] found that the audience’s behavior was affected by the excitement and suspense generated during watching. The excitement and suspense mainly came from two aspects: the expectation of the audience to win the game, and the curiosity about the game challenge results of the audience. Scully-Blaker et al. [22] explained the cause of suspense in video game live streaming. In pre-recorded media, the results might already be known. Compared to pre-recorded media, real-time performance of video game live streaming maintains the audience’s suspense through high and low moments, as well as results of the competition games. So, the attraction towards competition games will be enhances. Second, entertainment is the most basic service provided by the live streaming platform to its audience; however, the audience watching the live streaming is also influenced by their own entertainment motivation [9,29,33,38]. Through a questionnaire survey about the Twitch platform, Gros et al. [9] found that regardless of whether the audience spent money, entertainment was the main motivation to use video game live streaming, and that the more time the audience spent, the higher the entertainment motivation was. Therefore, entertainment motivation is speculated to be a significant factor in impacting audiences to become addicted to video game live streaming. However, the entertainment was not always positive, but also negative and hurtful, for example, when the audience criticized or spoofed streamers to entertain themselves [38]. Therefore, the entertainment by teasing the live streamer should also be considered in the affective demands that affect live streamer’s participation in the live streaming.

In addition to major factors that satisfy the five aforementioned demands, other influencing factors include the audience’s preference of content. Chen and Pan [32] analyzed the chat records of users and found that game content was an important factor that affected the audience to watch live streaming. Furthermore, the novelty of video game live streaming content would attract audiences [19]. In addition, although action games earned the lowest income in the live streaming, they were the most popular in terms of audience numbers and viewing duration [28]. Additionally, 10% of individual streamers accounted for 95% of the audience, from which it was observed that streaming popularity followed a highly skewed Pareto principle [11]. This also provides ideas for future research. By examining the characteristics of the most popular top 10% of live streamers, we can analyze the reasons why audiences want to participate in video game live streaming.

## 6. The User Interaction Behavior

Parasocial interaction, proposed by Horton and Richard Wohl [39], was an illusive sense. It refers to the audience’s mutual awareness and intimacy with media personas (e.g., celebrities, news hosts, and characters). Based on the quasi-social theory, the interaction between the streamer and the audience might be a two-way interaction mode, or a one-way or a one-to-many mode. However, when the audience scale exceeded a certain number, the one-way mode was more prominently observed [30]. In the following section, we focus on a two-way interaction between the streamer and the audience. We mainly discuss chat interaction, reward and subscription, and learning. In addition, we analyze the effect of three interactive behaviors on the streamer and the audience. Table A3 presents the effect factors, research theories, research methods, and sample data from eight literature that study in affecting interactive behaviors. There is a wealth of research methods involved in the researches of interactive behavior, not only questionnaires, interviews, and text mining of chat messages, but also laboratory experiments, case studies, as well as the literature first through laboratory experiments, then through questionnaires.

### 6.1. Chat Interaction

Live streaming is a medium combining real-time video and text chat [40,41], integrating players and audiences into a communication environment [11]. Wang and Li [42] showed that audiences’ commenting was mostly affected by the number of viewers, the gender of streamers, the number of likes, the number of gifts, and the duration of the live streaming. Nelson and Yasunobu [43] found that there was a significant positive correlation between the audience’s experience and perceived validity, and their active chatting behavior. When an audience’s question received a response from the streamer, the member would be more willing to keep chatting. Moreover, in special cases when the streamer replied to a unique message from an audience member, the latter would be more willing to follow the streamer [23]. By studying the user informational behavior (production, reception, reaction), Diwangjid et al. [44] found audiences are more likely to participate in the live streaming when streamers actively feed-back the message sent by audiences. Therefore, the chat interaction between audience and streamer would also positively affect the size of the audience.

### 6.2. Rewards and Subscription Interaction

Audiences are willing to invest not only time, but also money on the live streaming platform. Some audiences choose to pay a reward or subscribe to streamer by spending more money. Some scholars believe that these members can receive unique benefits from the streamer. Wulf et al. [23] and Gros et al. [9] mentioned that most streamers would immediately express their gratitude in words; in some cases, some streamers would also play songs to reflect their appreciation towards new rewarders and subscribers. As a result, audiences’ rewards and subscriptions were affected by timely feedback from streamers. Furthermore, Postigo et al. [41] found that not only would the rewarders and subscribers be mentioned, but they might also get the chance to individually chat and have a more intimate social experience with the streamer. However, Hamilton et al. [19] also proposed that the purpose of audience subscription was to reduce the noise of audiences gathering together, as subscribers would not receive spam messages when chatting. There is some controversy in existing research on the motivation of willingness to donate behavior. Audiences may be rewarded or subscribed, because they want to get the streamer attention, or they may want to reduce the noise.

Streamers, viewing duration and the number of audience members affect the audience’s intention to reward. Gros et al. [9] found that the audience’s reward intention was affected by the viewing duration and number of audience members. Specifically, the longer the audience watched at a single time, the higher the reward intention grew. Zhu et al. [45] analyzed relevant data of the Douyu live streaming platform and found that there was a positive correlation between the number of audience members and the reward times, and between the number of audience members and the amount of reward given. Chen and Pan [32] also agreed that the interaction between audience and streamer would affect the times of the audience watching live streaming and consequently their willingness to reward. Three of the eight papers collected mentioned influencing willingness to reward. Two of the articles analyzed the Twitch platform, and another one was Douyu. The research methods involved questionnaires and data mining, respectively.

Audiences giving rewards and gifts have the characteristics of time and inclination. Karhulahti et al. [38] made an individual analysis of e-sports streamer Ali Larsen and found that there would be fewer questions and rewards during the competition. As expected, the asking of questions and reward behaviors increased when the game mode changed to the question and answer session. Zhu et al. [45] also suggested that the distribution and presentation of gifts were inclined; most gifts were purchased by a few audience members, while most gifts were received by a few channels even though substantial numbers of audience members had gathered in these few channels.

### 6.3. Interactive Learning

Unlike the traditional learning model—where only the apprentice is learning—the existing literature suggests that learning in video game live streaming is involved on both sides. Video game live streaming facilitates learning interaction between the streamer and the audience. Streamers and audiences play roles of mentors and apprentices, respectively [18]. Wulf et al. [23] found, through an online questionnaire survey, that there was a special “learning” relationship between streamer and audience in which the streamer answered the audience’s questions about game skills and strategies through chat.For example, in the Speedrunner community, a streamer finds and takes advantage of the bugs in a game to teach the audience skills and strategies [24]. However, Payne et al. [21] showed that the learning interaction between professors and learners, and learners with other learners, could be improved. In addition, Greenberg [46] also found that a streamer’s skill could be affected by the number of followers (audiences, rewarders, subscribers) and the skill level of followers. For example, the communication between streamers and followers will typically contain skill information about the game, hence, the streamer’s skill will improve accordingly. Payne et al. [21] showed that video game live streaming, as a special learning approach, was responsive to all stakeholders and optimized various skills through mentorship (live streamers), learners (audiences), and collaborators (other audiences).

Streamer’s mentorship skills and learner’s character will affect the learning of the users on video game live streaming. Based on the cognitive load theory, Payne et al. [21] found that the teaching effect of novice instructors was at par with that of experts. Furthermore, some scholars have explored the effect of apprenticeship character and found that a feasible interaction between two learners could benefit extroverts. Predictably, learners with very amiable temperaments benefited more from novice instructors.

## 7. Platform Influence

The video game live streaming platform is an important factor influencing the participation of both audiences and streamers. Features of video game live streaming platforms include website attractiveness, gamification mechanisms, socialization, convenience, digitalization and so on, all of these features attract streamers and audiences to participate in live streaming. The following section reviews the impact of live gaming platforms on participation behaviors of live broadcasters and audiences.

### 7.1. The Effect on Streamers

The attractiveness of the website can impact streamers’ behavior, which includes the website’s graphic design, user interface, and live channel. These factors had a positive impact on the participation of streamers [8]. Gandolfi [28] analyzed the questionnaire survey of Twitch platform users and found that the effect of playing video and audio on the platform would affect the aesthetic feeling of the streamer, and indirectly affect the behavior of the streamer participation during the live streaming. The graphic design and user interface of the website gave users a better user experience and visual impact. Real-time live channels and user interaction interface design could better meet the social demand of users. Therefore, website attractiveness is an important factor affecting live streamer engagement.

The design of gamification mechanisms on the live streaming platform will affect the behavior of continuous live streaming. Gamification is the use of game design elements in non-game environments, products, and services to stimulate user expectations, such as points, badges, or money [15]. Sjoblom et al. [47] found that Twitch platform attracted potential streamers through gamification mechanisms, and further maintained the relationship between the platform and streamers with financial incentives, such as monetary rewards, which would motivate streamers to continue streaming.

In summary, features of video game live streaming platform would attract streamers to participate in live streaming such as website’s graphic design, user interface, and live channel, gamification mechanism. Therefore, these platform features mentioned above should be considered when studying streamers’ addiction to video game live streaming.

### 7.2. The Effect on Audiences

The design of gamification mechanisms and socialization can affect the audience’s participation in the live streaming. Subscriptions, rewards, and game mechanism design, along with the sociability of the chat room design on the platform can meet the audience’s social demands. Fietkiewicz and Stock [15] proposed that the design of gamification mechanisms strengthened the interaction between streamers and audiences, and between audiences with other audiences on such platforms. Vosmeer et al. [29] found that the audience’s social demands (interaction with other audiences) were an important factor that affected their participation. It was observed that the audience generally enjoyed the unique functions of the platform design, such as gamification mechanisms and chat rooms [23]. In addition, audiences who spent more money on the platform had a stronger social motivation, such as having the opportunity to enter an exclusive chat mode (where chatting with live streamers alone is enabled) through subscription [9].

The digital experience and convenience on the platform can also affect the audience participation. Neus et al. [48] used the Motivation Scale for Sport Consumption (MSSC) in a questionnaire survey on e-sports event consumption to investigate online and offline audiences. They found that the digital experience of online live streaming would affect the audience’s satisfaction. Comparing questionnaire survey responses of offline and online e-sports audiences, Taylor [13] found that audiences who watched online live streaming thought that this platform was more convenient. In addition, Hu et al. [30] found that compared with the live streaming of talent shows, audiences of video game live streaming identified with the platform, and this had a strong correlation with their willingness to continue watching.

The gamification mechanics and socialization of live streaming platforms meet the social demands of the audience. And the digital experience and convenience better satisfy the viewing experience of the audience. In conclusion, features of the live streaming platform would attract audiences to participate in the live streaming, including gamification mechanisms, socialization, digital experience and convenience. Therefore, these platform features mentioned above should be considered when studying the audience’s addiction to video game live streaming.

## 8. Discussion

Video game live streaming platforms are becoming more and more popular, with both streamers and audiences spending more time on these platforms. It has become a matter of concern which factors affect user participation in video game live streaming. This paper analyzed the existing literature in terms of both user demands and platform characteristics, and found that user demands in the five major aspects of social integration, personal integration, tension release, affection, and cognition affect user behavior in the live streaming. This provides a comprehensive analysis of these research current state about user engagement behavior in video game live streaming. Furthermore, there are related implications in terms of users indulging in video game live streaming.

As with game obsession, audience and streamer obsession with live gaming may be a mental health issue of concern. By summarizing the factors that impact the user behavior, this paper found these factors that might affect addiction to video game live streaming from two aspects, namely, user demands and platform impact. For streamers, social integration, personal integration, affective and additional rewards should be fully considered. For audiences, attention should be paid to affecting factors such as social integration, personal integration, tension release, affective demand, and cognition, and game category. Of these, social and affective demands are the more important impact factors for both streamers and audiences. And for cognitive demands, no relevant factors have been found to affect the behavior of streamers. The audience wants to learn more about game skill through the platform, but there is not much desire to satisfy the cognitive demands. Therefore, this paper argues that cognitive demand is unlikely to be a factor in user addiction to video game live streaming. In addition, platform features such as website attractiveness, gamification mechanisms, socialization, convenience, digitalization, and so on, should be highlighted in the factors of user addiction to video game live streaming.

### 8.1. Future Directions

Although some valuable results have been achieved in existing research, further research and improvement are expected due to the fact that user behavior in video game live streaming is relatively lacking in the field of mental health research. This paper proposes that future studies might explore the following aspects: First, expansion of the research platform. At present, more than half of the research on video game live streaming was based on the Twitch platform. Previous studies have found that existing characteristics of video game live streaming platforms would affect users’ participation behaviors. On this basis, research should be extended to other platforms—perhaps those in different countries—to explore the influence of personalized and differentiated designs of various platforms on users. In addition, most existing video game live streaming platforms were connected to other social platforms, and users could converse with each other on these platforms. For example, Douyu allows video link sharing, therefore, both audiences and streamers can share video links to WeChat, Weibo, QQ, and other social platforms. For the sake of convenience, users are also allowed to log in directly through such social media accounts. Hence, we can explore impact of this kind of cross-platform joint approach on users of video game live streaming in the future.

Second, expansion effect factor on streamer addiction. Existing literature is lacking on studying factors affecting the streamer participation. This article argues that the effect of tension release and cognitive on streamer participation should be considered. Streamers could also relax via streaming playing games by themselves. Because playing the game was recognized as a way of relieving stress and the live-streaming process allowed for conversation with the audience. In addition, it is also possible that cognitive demands of streamer could be fulfilled by streaming. Although existing research did not indicate that streamer was engaged in streaming to fulfill cognitive demand, and it was mentioned in the literature that the live streamer might also learn some gaming skills through interaction with audience during process of acting as a teacher [46]. Thus, future research could delve into tension release and cognitive demands among the factors of streamer participation. Furthermore, existing research on the impact of monetary incentives is controversial. The degree of monetary incentive makes a difference between amateur and professional streamers, and perhaps to some extent, money makes a greater incentive for amateur streamer. Because professional live-streamers will be at risk of losing their jobs [27], but amateur streamers are not.

Third, expansion in methods for acquiring research data. Internet data mining has become a major trend in the field of Internet behavior research. In future research, network data mining can be used to collect research data. Compared to general questionnaire surveys, interviews and other such methods, network data mining can obtain primary data more objectively and accurately. For example, Nascimento et al. [49] created a prediction model for the number of audience chat messages by scrutinizing Twitch platform data. However, it was limited to only acquiring relevant data of users’ chats. In the future, data mining can be used to acquire datum of other aspects of the platform for video game live streaming research. Furthermore, laboratory experiments should also be considered to examine the factors that impact user participation. Laboratory experiments can better control uncontrollable factors in the real environment and better explore the main effects. For example, to examine the efficacy of Twitch-based worked-example learning, Payne et al. implemented a laboratory experimental design [21].

Fourth, expansion user behavior of video game live streaming with the implementation of Tandem Play (i.e., when an audience member and streamer play a game together). At present, studies are largely based on users’ behavior under the condition that the streamer performs for the audience member. Scully-Blaker et al. [22] have defined Tandem Play in relation to video game live streaming in the existing literature and discussed the influence of various restrictions and functions of the Twitch platform on users’ behavior in this format. Based on this, future research can further explore users’ behavior in video game live streaming while implementing Tandem Play.

### 8.2. Limitations

Based on the existing literature, this paper comprehensively defined the video game live streaming, outlines the characteristics and the community classification of the video game live streaming. Moreover, we analyzed the effect factors on user participation and constructed the theoretical framework. But there are also some limitations in this paper. First, a limited number of papers—only 24—were collected. This might be due to the title and keyword filtering phase, which indicated that the study generally pertained to user behavior on the live streaming and online gaming. Since only English-language studies were included in the review, some key studies in other languages might be excluded. Second, we only analyzed user behavior in video game live streaming in terms of user demand and live streaming platform characteristics. However, the factors impacting users’ participation in video game live streaming are more complex and other aspects such as the type of game are not considered.

## 9. Conclusions

This paper is a systematic literature review on the impact factors of user engagement in video game live streaming. At present, no similar literature review is found in the field of users’ addictions to the video game live streaming. According to the 24 relevant literature references, we summarized the definition, features, community classification, and users of video game live streaming. Then, we analyzed these papers separately from factors of live streamer, audience, and interactive behavior. Among these, streamer interaction behavior includes chatting and learning. Audience interaction behavior includes chat interaction, reward and subscription, and learning interaction. From the perspective of the platform and two user groups, this paper constructed a framework of effect factors of user engagement in video game live streaming. It was found that effect factors on video game live streaming user behavior mainly involve two aspects: user demands and platform influence. Video game live streaming can meet user demands in social integration, personal integration, tension release, affection, and cognition. The website attractiveness, gamification mechanisms, socialization, convenience, digitalization and other features of the platform would attract users to participate in live streaming. We hope these findings can help address the mental health issues of video games live streaming addiction. Furthermore, this article summarized future research directions from existing research limitations such as research platform, effect factors, research methods, user behavior. This article hopes to provide ideas for future research.

## Figures and Tables

**Figure 1 ijerph-17-03328-f001:**
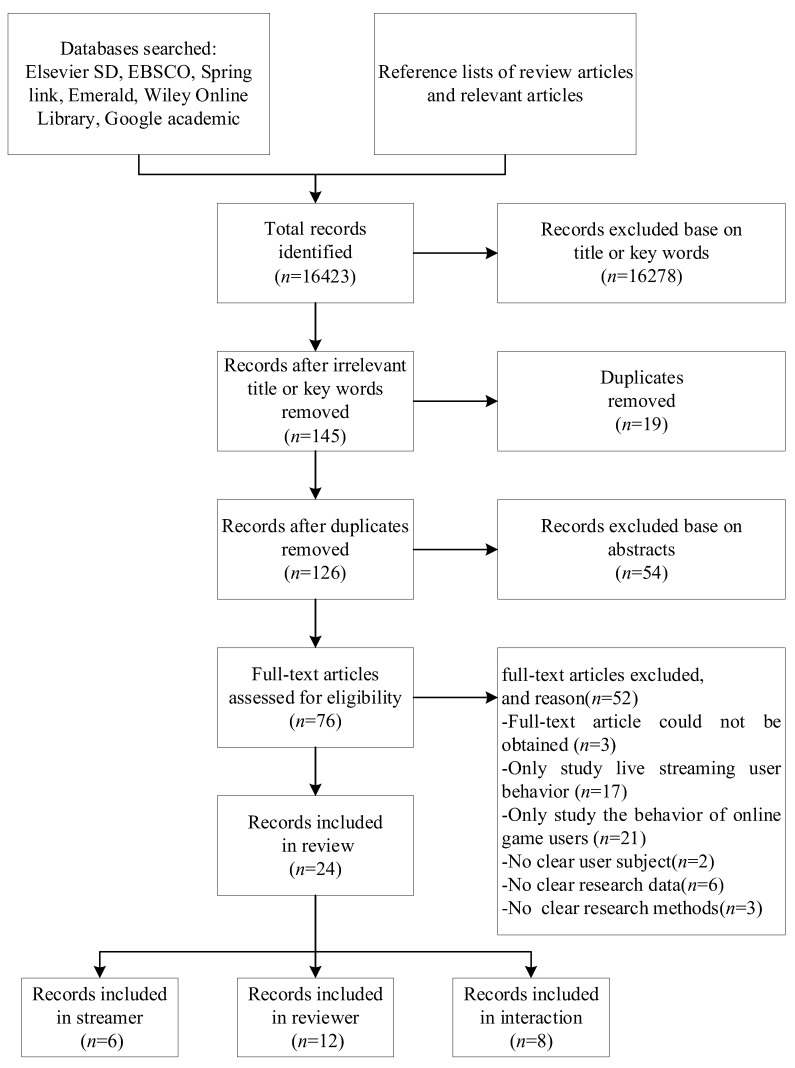
Flowchart of the systematic review process. Note: There were two instances in which the same paper was included for both records for the audiences and records for interaction.

**Figure 2 ijerph-17-03328-f002:**
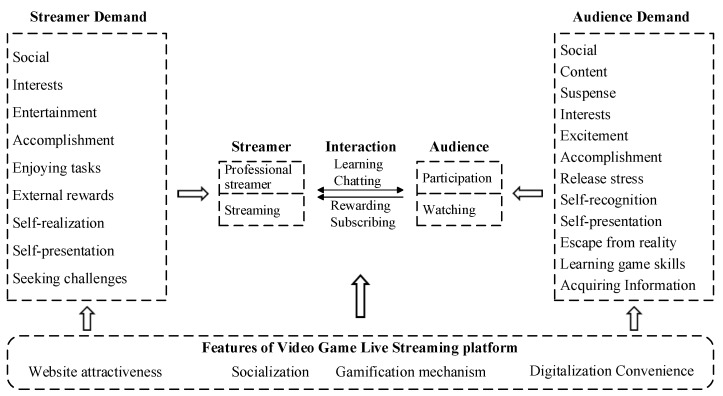
The research framework of video game live streaming. Source: Compiled by the authors.

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
