# Peer review of "A Systematic Review of Literature on User Behavior in Video Game Live Streaming"

_ijerph, 2020, doi:10.3390/ijerph17093328_

Round 1

Reviewer 1 Report

Dear Authors,

This paper presents

User Behavior in Video Game Live Streaming.

Detecting the inferiority is an interesting topic, but it's not as easy that you mentioned in your paper.

Here are some of the issues that you need to solve.

1- most of your references are not up to date, and you need to change them and do the broad research to find more updated references.

2- You didn't' mention why you chose twenty research literatures for analyzing and checking the data?

3- what is your customized method? You need to explain how it is working and how you chose it.

4- Understanding user behavior is not an easy task at all, and you need to divide your work based on the explicit and implicit inferiority of the text and participators in the video games.

5-You just reviewed some of the relevant works in this field without any research and it's not sufficient for publication in the journal. 

6- I am suggesting to revise your paper and improve the level of your language. 

7- Figure 2, The research framework of video game live streaming is not clear for the reader and I think you missed of the component in your presented model. 

Author Response

  1. most of your references are not up to date, and you need to change them and do the broad research to find more updated references.

Response: Thanks for reviewers’ comment and suggestions. We have tried our best to search the latest relevant literature. Nearly half of the 24 selected papers are from the last three years. Based on the screening rules established, we added four new references on researching into the factors affecting user behavior in video game live streaming.

One of them is from 2019 and the other three are from 2020. The following new literature were added.

[1] Chen, C. Y.; Chang, S. L.; Moderating effects of information-oriented versus escapism-oriented motivations on the relationship between psychological well-being and problematic use of video game live-streaming services. J. Behav. Addict. 2019, 8(3), 564–573. https://doi.org/10.1556/2006.8.2019.34

[2] Diwanji, V.; Reed, A.; Ferchaud, A.; Seibert, J.; Weinbrecht, V.; Sellers, N. Don’t just watch, join in: Exploring information behavior and copresence on Twitch. Comput. Hum. Behav. 2020, 105, N.PAG. https://doi.org/10.1016/j.chb.2019.106221

[3] Lim, J. S.; Choe, M.-J.; Zhang, J.; Noh, G.-Y. The role of wishful identification, emotional engagement, and parasocial relationships in repeated viewing of live-streaming games: A social cognitive theory perspective. Comput. Hum. Behav. 2020, 108, N.PAG. https://doi.org/10.1016/j.chb.2020.106327

[4] Wang, M.; Li, D. What motivates audience comments on live streaming platforms? PLoS ONE, 2020,15(4), 1–12. https://doi.org/10.1371/journal.pone.0231255

  1. You didn't mention why you chose twenty research literatures for analyzing and checking the data?

Response: We agree with the reviewer that reasons for the selection papers is inadequate. A more detailed description of the reasons for choosing literature has been added in Chapter 2. First, the main matter of the literature should be about user behavior and effect factors of video game live streaming on live streaming platforms, such as Twitch, YouTube Gaming, DouYu TV, Huya.com and another popular online game platform. Second, the papers must include clear live streaming user objects, research methods, research data. Third, Full-text paper could be obtained. The specific revisions can be found in Chapter 2, lines 85-108 from the text.

  1. what is your customized method? You need to explain how it is working and how you chose it.

Response: Following reviewer’s suggestion, we have detailed our process of selecting 24 pieces of literature in the first two paragraphs of the second section of the text. We described the keyword selection, search time, search database, and screening conditions respectively. Please read lines 73-94 of the text for specific revisions.

Understanding user behavior is not an easy task at all, and you need to divide your work based on the explicit and implicit inferiority of the text and participators in the video games.

Response: Thanks for reviewer’s advice. The original intention of the paper is not to study the inferiority. It may be that our use of words has led to misunderstanding. Now we have corrected the use of words. If there is any problem, please point out. We are willing to continue to modify.

  1. You just reviewed some of the relevant works in this field without any research and it's not sufficient for publication in the journal. 

Response: Thanks for reviewer’s comments about the literature review. we have made the following revisions. First, the revision focuses on the analysis of each broad category of factors affecting user behavior in terms of research theory, research methods, research data, etc. Second, the framework diagrams and literature tables of the factors affecting the behavior of live game users are combined with the existing literature to analyze the three subjects of streamer, audience and platform. Specific changes are highlighted in chapters IV to XII of the text.

Furthermore, through an analysis of the existing literature, this paper summarized the findings of the study and the implications for the study of user addiction to video game live streaming in the Discuss section. It also focuses on the future direction of research on user participation in video game live streaming.

  1. I am suggesting to revise your paper and improve the level of your language.                                                 

Response: Thank for these comments and suggestions. We have implemented all the minor improvement suggestions in the new version of the manuscript. The manuscript has also been reviewed by a native English speaker to check for grammar, language and style.

  1.  Figure 2, The research framework of video game live streaming is not clear for the reader and I think you missed of the component in your presented model. 

Response: Thank for pointing out the problem in our Figure 2, which we have modified as follows. First, we add subheadings to each section to clarify the content of each section. Second, In the interactive behavior section, we make clear that learning and chatting are two-way interactive behaviors by using double arrows, and rewarding and subscription behaviors are one-way behaviors by using single arrow.

Reviewer 2 Report

The English needs substantial work because in parts grammatical and syntactic errors limits ease of reading.  The work is a fairly pedestrian literature review (and does not pretend to be otherwise - which is good).  The methodology is clearly explained but perhaps too much of the papaer was devoted to this aspect.

A suggestion could be to restructure the good material in the introduction so that the analysis of the selected papers are analysed within that frame of reference and then the strengths and weaknesses of the methodologies used in the references can be considered more clearly.  This is a very interesting topic, so clarity of writing matters.

The two important sections - the discussion and the conclusions are much too short.  The limitations in the methodology are acknowledged, and it would be of interest to have more discussion on this - especially more suggestions on better methodologies for investigating emerging topics of this nature.  The suggestions for future research are very useful and the most important part of the paper. This is an area where a lot more work would be appreciated, because the ideas are not always as well articulated as they could be. There is a hint that the authors are aware that not all the selected papers were adequately analysed.

Author Response

  1. The English needs substantial work because in parts grammatical and syntactic errors limits ease of reading. 

Response: Thank for these comments and suggestions. We have implemented all the minor improvement suggestions in the new version of the manuscript. The manuscript has also been reviewed by a native English speaker to check for grammar, language and style.

  1. The work is a fairly pedestrian literature review (and does not pretend to be otherwise - which is good).  The methodology is clearly explained but perhaps too much of the papaer was devoted to this aspect.

Response:We thank the reviewer for this comment. Following his/her recommendation,in this revision, we focus on adding an analysis of the existing literature in terms of research methods, research theory, research data, and consistency between existing findings. Specific revisions are highlighted in Chapters IV to XII of the text.

  1. A suggestion could be to restructure the good material in the introduction so that the analysis of the selected papers are analyzed within that frame of reference and then the strengths and weaknesses of the methodologies used in the references can be considered more clearly. This is a very interesting topic, so clarity of writing matters.

Response:Based on reviewer suggestions, we have made revisions to the introduction. We added game addiction to the first paragraph of the introduction, which will lead to the study of live game addiction. In the second paragraph, we added the impact of the platform on user engagement behavior. So it can be seen clearly that the analysis of this paper is the four aspects of live streamer demand, audience demand, the impact of interaction behavior, and platform impact. The last paragraph briefly describes the analytical thinking of the paper. Specific revisions are highlighted in the introduction.

  1. The two important sections - the discussion and the conclusions are much too short. 

Response: Thanks for reviewer’s comments on the discussion and conclusions. We've made a major revision to these two parts. In this revision, we have added new content to these two sections. First, we added a section at the beginning of the discussion that analyzes the preceding text. Secondly, a sub-section has been added to the discussion section - a limitation of this article. Then, in the future directions section, an explore effect factor on streamer addiction was added. Finally, laboratory experiments were added to the research methods in outlook section. For specific additions, see the highlights of the discussion section.

Revisions to the conclusion section recapitulate the full text and summarize the findings and results of this literature review. Finally, the author's expectations were clarified for this paper.

  1. The limitations in the methodology are acknowledged, and it would be of interest to have more discussion on this - especially more suggestions on better methodologies for investigating emerging topics of this nature.  The suggestions for future research are very useful and the most important part of the paper. This is an area where a lot more work would be appreciated, because the ideas are not always as well articulated as they could be.

Response: Based on reviewer’s suggestions, in the revisions to the discussion section, we have added a proposed research methodology for the study of user engagement behavior in video game live streaming, added laboratory studies other than data mining, and described the superiority of this research methodology in such studies.

  1. There is a hint that the authors are aware that not all the selected papers were adequately analysed.

Response: We agree with reviewer that we have not conducted an adequate analysis of the collected literature. Therefore, in this revision we combine the three tables of the collated audience, live streamers, and interactive behavioral impact factors for further analysis in terms of research theory research methods, research data, etc. Then, we were reflecting on the psychological factors that affect users' engagement behavior, and analyzed the factors that impact users' indulgence in video game live streaming. Specific revisions are highlighted in chapters IV to XII of the text.

Reviewer 3 Report

This paper is a systematic review examining the determinants of participation of users including streamers and audiences in video game live streaming. The purpose of the review is to provide guidance to enterprises to enhance participation. The review is well organized and well written.
The following issues are raised:
1. The relevance of this work to readers of a public health journal was unclear.
2. The discussion should have compared the current findings to the existing literature. Was anything new revealed from this systematic review?
3. The authors followed PRISMA guidelines but did not discuss their process of data collection and whether individual studies were assessed for risk of bias.
4. Under the discussion of “Chat Interaction”, the authors indicated that interactions between streamers and the audience will affect audience size but as written it was unclear whether interactions are related to increased or decreased audience size.

Author Response

This paper is a systematic review examining the determinants of participation of users including streamers and audiences in video game live streaming. The purpose of the review is to provide guidance to enterprises to enhance participation. The review is well organized and well written.
The following issues are raised:

  1. The relevance of this work to readers of a public health journal was unclear.

Response: Thanks to the reviewer for the comments. The purpose of writing this paper is revised in the abstract, citation and conclusion sections. For Internet media such as video game live streaming, indulgence is an important factor affecting mental health. To understand the psychological demands of users using live streams, and to provide reference for solving psychological problems such as indulgence to live streaming. This paper explores the factors that impact users' behavior in participating in video game live streaming, focusing on the psychological needs of users of live games, as related to the journal's mental health column.

  1. The discussion should have compared the current findings to the existing literature. Was anything new revealed from this systematic review?

Response: Thanks for reviewer’s suggestions for the discussion section. Following her/his suggestion, we have added to the discussion section a comparison between the findings of this literature review and the existing literature, and summarized the findings of this literature review in the conclusion section. For specific modifications, see the discussion section on painting highlights.

  1. The authors followed PRISMA guidelines but did not discuss their process of data collection and whether individual studies were assessed for risk of bias.

Response: Agreeing with the reviewer, we did not have a detailed literature selection process. we have detailed our process of selecting 24 pieces of literature in the first two paragraphs of the second section of the text. We describe the keyword selection, search time, search database, and screening conditions, respectively. Please read lines 73-94 of the text for specific changes. As far as possible, we developed screening conditions to avoid literature selection errors resulting from personal assessment bias.

  1. Under the discussion of “Chat Interaction”, the authors indicated that interactions between streamers and the audience will affect audience size but as written it was unclear whether interactions are related to increased or decreased audience size.

Response: Thanks to the reviewer for the comments. In this revision, we cite the most recent search literature in which it is explicitly suggested that chat interaction between live streamers and audiences increases the size of the audience. Please see revised text in lines 406-409 of the text

Round 2

Reviewer 1 Report

Thanks for applying most of the concerns!

Reviewer 2 Report

The revisions vastly improve the paper - done extremely quickly and adequately.  They noted that a native speaker of English went through and corrected the English, but there are still errors in the use of articles and tenses - but these do not materially affect readability - but would be useful to fix these anyway - a couple of typos as well.

Reviewer 3 Report

The authors have addressed the reviewers concerns. The use of the English language and style needs to be re-checked.